# Correlation between Morphological Characteristics of Macular Edema and Visual Acuity in Young Patients with Idiopathic Intermediate Uveitis

**DOI:** 10.3390/medicina59030529

**Published:** 2023-03-08

**Authors:** Ludovico Iannetti, Fabio Scarinci, Ludovico Alisi, Marta Armentano, Lorenzo Sampalmieri, Maurizio La Cava, Magda Gharbiya

**Affiliations:** 1Ophthalmology Unit, Head and Neck Department, Policlinico Umberto I University Hospital, Sapienza University of Rome, 00161 Rome, Italy; 2IRCCS Fondazione Bietti, 00199 Rome, Italy; 3Department of Sense Organs, Sapienza University of Rome, 00185 Rome, Italy

**Keywords:** intermediate uveitis, macular edema, optical coherence tomography, visual acuity, retinal morphology

## Abstract

*Background and Objectives:* Macular edema (ME) is a common complication of intermediate uveitis (IU). It is often responsible for a decrease in visual acuity (VA). Three distinct patterns of macular edema have been described in intermediate uveitis, namely, cystoid macular edema (CME), diffuse macular edema (DME), and serous retinal detachment (SRD). The current study aims to describe the characteristics of macular edema in young patients with idiopathic intermediate uveitis and to correlate its features with VA using spectral domain optical coherence tomography (SD-OCT). *Materials and Methods:* A total of 27 eyes from 18 patients with idiopathic IU complicated by ME were included in this retrospective study. All patients underwent SD-OCT; data were gathered at the onset of ME. Best-corrected VA (BCVA) was correlated with the morphological features of ME. *Results:* BCVA was negatively correlated with Ellipsoid Zone (EZ) disruption (*p* = 0.00021), cystoid pattern (*p* = 0.00021), central subfield thickness (CST) (*p* < 0.001), and serous retinal detachment (0.037). *Conclusions:* In ME secondary to idiopathic IU, VA negatively correlates with Ellipsoid Zone disruption and increases in CST. Moreover, vision is influenced by the presence of cysts in the inner nuclear and outer nuclear layers and by the neuroepithelium detachment.

## 1. Introduction

Intermediate uveitis (IU) represents the widest of four major categories of uveitis proposed by the International Uveitis Study Group (IUSG). IU predominantly affects patients under the age of 40 and affects approximately 10% of the general uveitis population. It accounts for 8–18% of uveitis cases and up to 38% of uveitis in the pediatric population [1,2]. Usually, the disease presents as bilateral in 70–90% of cases. No difference in gender prevalence has been observed [3]. Although the disease does not present itself as hereditary, a certain common HLA recurrence in some families, such as HLA-A*28 HLA-DRB1*15, HLA-B*51, and B*08, has been observed [4]. Moreover, Tang et al. observed that patients with HLA-DR15-related IU showed a tendency to develop the concomitant systemic findings of other related disorders, such as multiple sclerosis, optic neuritis, and narcolepsy [5]. In addition, a concomitant genetic predisposition may act in concert with an exogenous infection to determine the development of the disease. For instance, it has been suggested that HLA class II protein may act as a cofactor during the infection of B lymphocytes by the Epstein–Barr virus. The Epstein–Barr virus has been linked to the development of multiple sclerosis. Some authors suggest that there may be a common genetic predisposition in patients affected by multiple sclerosis and the Epstein–Barr virus [6]. From an etiopathogenetic point of view, several hypotheses have been made in the previous years. Gartner suggested that the remnants of the hyaloid may act as an immunogenic stimulus in pars planitis, while others suggested breakdown of the blood–retinal barrier as the first inflammatory trigger [6]. The precise etiology of IU is not known, although it is oftentimes associated with systemic conditions such as sarcoidosis, multiple sclerosis, and several infectious diseases. IU is a chronic intraocular inflammatory disorder in which the vitreous represents the major site of inflammation [7]. Usually, the patient reports floaters and a minimally decreased vision. Clinical signs are represented by a minimal involvement of the anterior chamber; posterior synechiae, if present, usually involves the inferior iris. Vitreitis is a cardinal feature of the IU and is associated with snowballs and snowbanks. Snowballs represent inflammatory aggregates that can be found in the inferior peripheral vitreous. Snowbanks are inferior exudates of the pars plana and are usually associated with a more aggressive form of the disease [3]. 

IU is frequently related with peripheral retinal phlebitis with a percentage ranging between 16 and 36%. The persistence of the phlebitis may determine the formation of cyclitic membranes and neovasis. In a small percentage of cases, a retinal detachment can develop after an acute IU, both exudative and tractional [3,8].

Macular edema (ME) is characterized by a retinal thickening in the macular zone due to blood–retinal barrier (BRB) breakdown. Extracellular fluid accumulates in the intraretinal area or in the subretinal space. Inflammatory ME may be secondary to anterior, intermediate, posterior, or diffuse uveitis and is the main condition associated with vision loss in uveitis. The main cause of macular thickening in inflammatory conditions is inflammatory ME. Moreover, other causes can determine macular thickening in ocular inflammatory conditions, such as vitreo-macular traction from inflammatory epiretinal membrane; inflammatory choroidal vascularization; an association with papillary edema; or central serous chorioretinopathy due to the chronic use of steroid therapy. Inflammatory ME is due to the breakdown of the blood–retinal barrier, which is mainly formed of tight junctions between the endothelial cells of non-fenestrated capillaries and the retinal pigment epithelial cells [9,10,11].

Several molecular factors are involved in the inner BRB breakdown, such as VEGF; pro-inflammatory cytokines such as TNF-α, IL-1, TGF-β, and angiotensin II; as well as adenosine, histamine, and glucose [12]. In IU, increased levels of IL-6 and IL-8 in the aqueous humor were detected [13].

IU is characterized by a consistent prevalence of ME. In 21–52% of cases, ME is clinically significant, representing the most common cause of decreased visual acuity (VA) [14]. Longstanding ME can lead to retinal degeneration with permanent visual loss over time [15,16]. 

Optical coherence tomography (OCT) represents the gold standard technique for the diagnosis of ME, since it is non-invasive, reproducible, and sensitive. It quantifies the retinal macular thickness using mapping and may show fluid accumulation either at the inner or outer plexiform layer, a non-uniform photoreceptor outer/inner segment line, the presence of hyperreflective dots in the subretinal fluid, epimacular membranes, and vitreomacular traction. OCT is an imaging technique that allows the obtainment of images of the retinal layers with high detail [17]. The first available OCT technology was Time Domain OCT (TD-OCT). Subsequently, Spectral Domain OCT (SD-OCT) took its place and represents the most common model in use today [18]. Its high sensitivity makes SD-OCT suitable for detecting and monitoring uveitic ME, as it is able to discern forms of ME that may be otherwise undetectable by less-advanced methods [19]. Moreover, OCT delivers fundamental elements to study fluid deployment and organization, as well as revealing the morphology of the vitreo–retinal interface [16,18].The currently available SD-OCT instruments ensure an excellent display of the external limiting membrane (ELM); the photoreceptor inner segment—internationally referred to as the Ellipsoid Zone (EZ) and traditionally ascribed to the junction of the inner/outer segment (OS) of the photoreceptors; the interdigitation of the photoreceptor OS and the retina pigment epithelium (RPE), or OS/RPE junction; and the RPE–choriocapillaris complex. The OS/RPE junction line is also known as the cone outer segment tip (COST) line and it consists in a fine layer interposed between the EZ and the RPE [20,21].

Three different patterns of fluid distribution in uveitic ME have been identified: cystoid macular edema (CME), diffuse macular edema (DME), and serous retinal detachment (SRD) [22,23]. VA decreases with increasing fluorescein leakage and central thickness. A correlation has been established between the central thickness measured by OCT and VA. This correlation showed significant differences depending on the OCT pattern and was strongly dependent on the presence of CME. A negative correlation between central foveal thickness and VA has also been described by other authors [22,24,25]. DME was found to be associated with poor visual recovery after treatment, mainly because of the persistence of this edema presentation even after treatment [24]. On the other hand, several studies point to a negative visual outcome in patients with uveitic CME [15]. In a previous study, our group demonstrated in a mixed group of uveitic patients that visual acuity was negatively affected by CME and disruption of the photoreceptors [16]. Lardenoye et al. demonstrated that the cystoid pattern was associated with worse VA, advanced age, and chronic inflammation. The development of blindness in IU was caused, in most cases. by CME [11].

The purpose of the present study is to describe the OCT characteristics of ME at its first presentation in young patients affected by idiopathic IU and to correlate them with VA to better understand the impact of ME on the final functional outcome.

## 2. Materials and Methods

A retrospective study was carried out on patients younger than 25 years old affected by idiopathic IU complicated by ME at its first diagnosis from January 2020 to September 2022. The diagnostic suspect was first based on clinical evaluation. Exclusion criteria were patients affected by IU aged ≥ 25; IU associated with sarcoidosis, multiple sclerosis and infectious diseases, and other causes of hypovisus (e.g., amblyopia, optic nerve atrophy, retinal pathologies with macular involvement); media opacity and drug-induced ME. We considered several factors including age, sex, specific diagnosis, duration of the uveitis, and clinical characteristics. The evaluation of inflammation gravity and disease activity was conducted in accordance with the guidelines provided by the Standardization for Uveitis Nomenclature (SUN) Working Group [17]. Best-corrected visual acuity (BCVA) was assessed at first ME presentation with 5 m Snellen charts.

OCT scans (linear, radial, and volumetric) were performed to determine the retinal thickness and ME characteristics with an SD-OCT Spectralis OCT system (Heidelberg Engineering, Heidelberg, Germany). OCT images of insufficient quality determined the exclusion of the patient from the study. 

A correlation between BCVA and the following OCT parameters of ME was performed: pattern of ME (cystoid or diffuse) (Figure 1a,b), presence or absence of SRD (Figure 1c,d), presence or absence of foveal bulge (Figure 2a), presence or absence of subfoveal bubbles (Figure 2b), integrity or disruption of COST line (Figure 2c), integrity or disruption of Ellipsoid Zone (Figure 2d), integrity or disruption of ELM, central subfield thickness (CST). 

We decided to include only the patients aged < 25 and ME at its first diagnosis to investigate as homogeneous a sample of young patients as possible and to avoid the potential effect of longstanding or chronic ME on its OCT morphological features and on BCVA.

All patients received corticosteroid therapy administrated both topically and systemically to reduce inflammatory manifestations. In all patients, ME resolved after steroid treatment (systemic or peribulbar administration). 

This study followed the tenets of the Declaration of Helsinki. Informed consent was obtained from all subjects after an explanation of the nature and possible consequences of the study. In the case of minors, informed consent was obtained from the parents or legal tutors. 

### Statistical Analysis

The graphical analysis of population distribution for BCVA resulted in a non-normal distribution; therefore, we reported the median value. Metric variables are reported as mean (±SD). For the multivariate analysis of BCVA, cumulative link modeling (CLM in the ordinal package) was used in the “R for statistical computing” environment, version 2.15.2. [26,27] Graphical evaluation identified the data outliers, which were eliminated or used to establish collinearity. For the final BCVA, a logic link function with a flexible threshold was used. The full model was then evaluated by applying a stepwise procedure in both directions, automatically and manually, to select significative covariates. We reported regression coefficients and cumulative odds ratio. P-value was calculated with a wale test, and we considered significant values for *p* < 0.005.

## 3. Results

A total of 27 eyes from 18 patients (27 eyes) with ME associated with IU (8 males and 10 females), with a median age of 18 years (range: 13–23 years) and a mean follow-up of 46 months (range 15–37), were enrolled in the study. The median decimal BCVA was 0.8 (range: 0.55–0.95). The grade for vitreous cells and haze was 0.5+ in 5 eyes, 1+ in 13 eyes, 2+ in 9 eyes. 

The OCT’s morphological features in relation to the corresponding median BCVA are reported in Table 1. The SRD was observed in 13 eyes (48,1%) and always associated with the two main patterns (seven with CME and six with DME). Other morphological features observed were subfoveal bubbles in 12 eyes (44.4%), EZ disruption in 3 eyes (11.1%), COST line disruption in 6 eyes (22.2%), and ERMs in 17 eyes (62.0%). Average central subfield thickness was 459 µm (±153 µm) and central perifoveal thickness was 443 µm (±153 µm).

BCVA was negatively correlated with Ellipsoid Zone disruption (regression coefficient −5.6, *p* = 0.00021), cystoid pattern (regression coefficient −4.4, *p* = 0.00021), CST (regression coefficient −2.2, *p* < 0.001) and SRD (regression coefficient −1.9, *p* = 0.037). The significant correlations between VA and morphological characteristics are summarized in Table 2.

## 4. Discussion

In patients affected by ME secondary to idiopathic IU, the assistance of SD-OCT images can be a valuable help to the clinician in the characterization of edema features. In accordance with the literature, in this study, we recognized three principal patterns of edema: CME and DME, associated or not with SRD. Previously published studies identified comparable characteristics in patients suffering from intraocular inflammation and in diabetic macular edema [23,28,29,30]. 

Among our patients, the most common presentation was CME (66.6%) whereas DME was observed in 33.3% of cases. Interestingly, both CME and DME were found to be crucial for the development of SRD. In fact, these two distributions of ME showed a similar correlation with the risk of SRD formation, in accordance with the available literature [19,22,23,30]. 

In the present study, the main features affecting vision during inflammatory ME in idiopathic IU are EZ disruption, cystoid pattern, CST, and SRD. On the other hand, DME, the presence or absence of a foveal bulge, subfoveal bubbles, and COST line disruption do not affect VA. These results agree with those observed in our previous study performed on ME secondary to all types of uveitis [16].

The EZ can be recognized as a hyper-reflective line below the external limiting membrane and its integrity correlates with VA preservation during the course of ME [31]. Maheshwary et al. highlighted the role of the EZ in the maintenance of a good BCVA in patients with diabetic ME. They described a reduction in VA, measured in ETDRS letters, depending on the amount of Ellipsoid Zone loss. They hypothesized that the assessment of the EZ using SD-OCT images could be a predictive factor for VA in patients with diabetic ME [20]. Oster et al. observed the same correlation in ME caused by macular pucker. These findings underline the important connection between EZ integrity and visual performance [32].

In our study, among all the features analyzed with the multivariate regression analysis, EZ disruption appears to be the most influential factor negatively affecting VA. 

We did not find any significant correlation between COST line status and BCVA, although recent studies showed a negative correlation between COST line disruption and visual recovery in epi-retinal membranes (ERM), and between macular hole surgery and diabetic macular edema [33,34,35,36]. 

On the other hand, in a previous study, we found a negative correlation between the interdigitation zone disruption and VA in patients with uveitic macular edema [37]. Roesel et al. studied the characteristics of the junction between the inner and outer segments of the photoreceptors with the aid of fundus autofluorescence (FAF) and OCT in patients with ME. They correlated the anatomical data concerning the alteration and disruption of this retinal layer with the functional data evaluated with BCVA. They concluded that an increased central FAF, the presence of cystoid changes, a disrupted EZ, and ERMs were associated with poor vision [38].

Akduman et al. evaluated the modification of macular thickness in correlation with clinical inflammatory parameters such as cells and flare. They noticed an interdependence between macular thickness and VA, and between macular thickness and inflammatory indexes. Moreover, they highlighted that macular thickness measured with OCT could be a good predictive factor for VA in uveitic patients. They also suggested that objective OCT morphological characteristics may be more strictly correlated with VA than subjective characteristics such as flare or cells [39].

The Multicenter Uveitis Steroid Treatment (MUST) Trial investigated the BCVA of 479 intermediate, posterior, and panuveitic eyes at baseline and after a 2-year follow-up. A longer duration of uveitis, AC flare, the presence of cataract, pseudophakia at baseline, the persistence of vitreous haze, and macular thickening were associated with BCVA worsening [40].

Niederer et al. reported that early CME in IU can often be treated with a good improvement in VA in more than two-thirds of cases, while its sequelae as macular atrophy and scarring are significantly associated with moderate and severe vision loss [41]. These findings are in accordance with the MUST Trial and Follow-up Study, in which participants’ eyes with incident, persistent, or relapsed uveitic ME had consistently worse BCVA than eyes with resolved ME over seven years [42].

The influence of SRD on VA is still controversial. 

Separation of the neurosensory retina from the RPE leads to the deprivation of nutrition and oxygen supplies to the outer retina, which causes photoreceptor apoptosis and visual loss [43,44]. Successful reattachment of the neurosensory retina is crucial for vision recovery. Disruption of the EZ has also been observed in patients with retinal detachment using OCT and a better visual recovery is usually closely correlated with an intact EZ at the fovea after retinal reattachment. Retinal reattachment permits the restoration of the blood supply to the outer retina and the regeneration of the photoreceptors and the patients’ visual acuity is recovered subsequently [45,46]. SRD is also commonly observed in other macular disorders such as central serous chorioretinopathy and diabetic macular edema [47]. 

In a previous paper, we reported that VA is negatively correlated with SRD [16]. However, some authors report an ambiguous role for SRD in the deterioration of VA. More specifically, they report that SRD does not worsen the visual outcome in uveitic patients with ME [22,24]. Lehpamer et al. evaluated the influence of subretinal fluid on VA, based on OCT images during the course of inflammatory ME. They reported that the progressive augmentation of macular thickness due to subretinal fluid corresponded to a decline in VA at presentation. Moreover, the authors investigated the response of SRD and ME to uveitis treatment at 3 and 6 months and obtained an improvement in BCVA relative to the reduction in fluid accumulation. The resolution of ME was more consistent in eyes without SRD, although both eyes with and without SRD achieved similar levels of VA [48].

Weldy et al. also reported that uveitic eyes with subretinal fluid at presentation show worse initial VA when compared to eyes without SRF. However, they obtained a similar VA and CST in the follow-up after treatment [49]. Tran et al. described that DME had a worse visual prognosis compared to SRD with or without associated CME in uveitic patients [50].

Given the median decimal BCVA of 0.8, our study shows a limitation by not including uveitic eyes with macular edema and worse VA. Since patients with this condition can show a worse BCVA than 0.8 [11], our results might not represent the overall spectrum of the disease and might not be generalizable to more severely affected eyes.

Unlike other studies on similar subjects that reported larger samples but included different types of uveitis and a larger age spectrum, we focused our attention only on young patients affected by idiopathic intermediate uveitis [23,30,32]. 

The limit of the present study is the small sample of the study but this is mainly due to the choice of inclusion criteria—age (<25), type of uveitis (idiopathic intermediate), and ME (at its first diagnosis)—which were very strict to make the study sample as homogeneous as possible. 

## 5. Conclusions 

In conclusion, this study promotes the role of SD-OCT to gain information about retinal morphology and to correlate the images obtained with the visual data in patients suffering from idiopathic IU. We described two main patterns of fluid accumulation, cystoid and diffuse, which could determine the elevation of the neuroretina. Characteristics such as Ellipsoid Zone disruption, cystoid pattern, SRD, and central retinal thickening appear to be strongly associated with poor vision. Among these, the most influential features appear to be EZ disruption and cystoid form. Further studies are warranted to determine the long-term outcomes of the different ME patterns and the potential impact of different therapeutic approaches on these patterns and VA. 

## Figures and Tables

**Figure 1 medicina-59-00529-f001:**
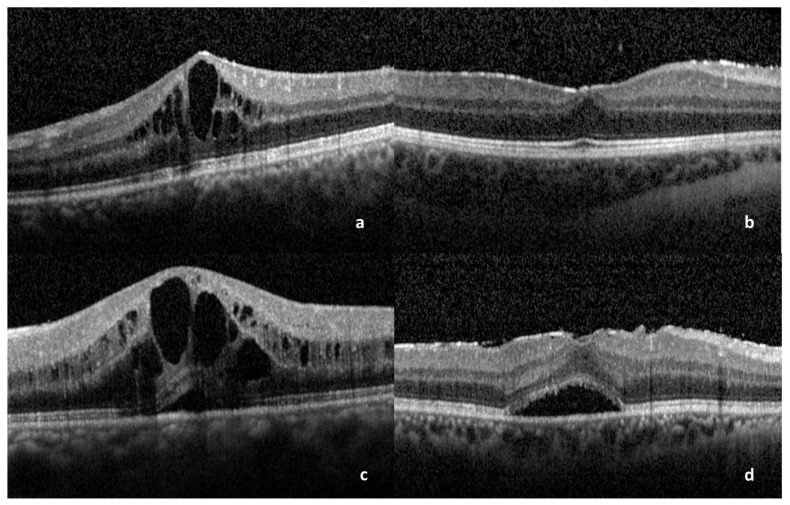
OCT scans showing: (**a**) Cystoid ME. (**b**) Diffuse ME. (**c**) SRD associated with cystoid ME. (**d**) SRD associated with diffuse ME. ME: macular edema, SRD: serous retinal detachment.

**Figure 2 medicina-59-00529-f002:**
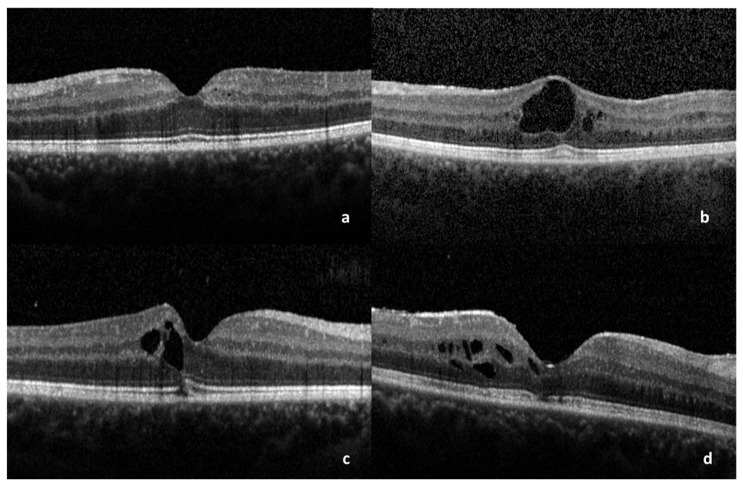
OCT scans showing: (**a**) ME with the maintenance of foveal bulge. (**b**) Presence of subfoveal bubble. (**c**) Disruption of COST line. (**d**) Disruption of EZ photoreceptor junction. ME: macular edema, COST: cone outer segment tip, EZ: Ellipsoid Zone.

**Table 1 medicina-59-00529-t001:** Results of the morphological features on OCT with the corresponding BCVA. DME: diffuse macular edema, CME: cystoid macular edema, SRD: serous retinal detachment, COST: cone outer segment tip, EZ: inner segment/outer segment, ELM: external limiting membrane.

	Number of Eyes (%)	Median BCVA (1st–3rd Quartile)
Total	27	0.8 (0.55–0.95)
DME	9 (33.3%)	0.8 (0.45–0.975)
CME	18 (66.6%)	0.9 (0.7–0.9)
SRD *	13 (48.1%)	0.7 (0.4–0.8)
Subfoveal bubble *	12 (44.4%)	0.8 (0.375–0.9)
Foveal Bulge *	11 (40.7%)	0.9 (0.8–0.95)
EZ disruption *	3 (11.1%)	0.6 (0.45–0.75)
ELM disruption *	0 (0%)	
COST line disruption *	6 (22.2%)	0.9 (0.675–0.9)
ERM *	17 (62%)	0.8 (0.6–1.0)
Central subfield thickness	459 (±153)	
Central perifoveal thickness	443 (±189)	

* Morphological features associated with the main groups DME or CME.

**Table 2 medicina-59-00529-t002:** The significant correlations between VA and morphological characteristics. CME: cystoid macular edema, SRD: serous retinal detachment, CST: central subfield thickness, EZ: Ellipsoid Zone.

Covariate	β Coefficients	Cumulative OR (95% CI)	*p* Value (Chi^2^ Test)
EZ disruption	−5.6 (−9.5–−2.5)	0.004 (7.83 × 10^−5^–0.08)	0.00021
CME	−4.4 (−7.4–−1.9)	0.012 (0.001–0.142)	0.00021
CST	−2.2 (−3.6–−0.19)	0.106 (0.028–0.304)	3.16 × 10^−6^
SRD	−1.9 (−3.9–−0.12)	0.146 (0.021–0.887)	0.037

## Data Availability

The data that support the findings of this study are not publicly available to protect the privacy of research participants but are available from the corresponding author L.I.

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
