# Peer review of "Correlation between Morphological Characteristics of Macular Edema and Visual Acuity in Young Patients with Idiopathic Intermediate Uveitis"

_medicina, 2023, doi:10.3390/medicina59030529_

Round 1
Reviewer 1 Report
Authors studied 3 morphological variants of macular edema in intermediate uveitis using OCT scan and correlated them with BCVA.
Authors should clarify the reason for age criteria <25 years.
Exclusion criteria- media opacity, drug induced CME could be part of exclusion criteria.
Patients may have a combination of each of the morphologic variants described (CME, DME, SRD, subfoveal bubble, COST, EZ disruption, etc.) Did authors exclude these cases?
Authors should mention in the methods if they evaluated BCVA at the presentation or at the resolution.
Authors should also mention the grading of vitreous haze in study eyes.
Please use consistent terminology “IS/OS interruption” or “EZ disruption”.
In the discussion, authors should emphasize on novelty in their study compared to previous reports.
Reviewer 2 Report
The current study aims to describe the characteristics 15 of macular edema in young patients with idiopathic intermediate uveitis and to correlate its features with visual acuity using spectral-domain Optical Coherence Tomography (SD-OCT).
they showed that ME secondary to idiopathic IU visual acuity negatively correlates with Ellipsoid zone interruption and increase of CST. Also, vision is influenced by the presence of cysts in the inner 24 nuclear and outer nuclear layers and by the neuroepithelium detachment. Although this is a well-written well-presented study, some drawbacks prohibit me to accept the article for publication.
1- The novelty of the current study is not enough.
2- The number of cases is low.
3- Discussion: “In our study among all the features analyzed with the multivariate regression, the Ellipsoid Zone interruption appears to be the most influential factor negatively affecting the 163 VA.” I cannot find a multivariate regression in the results section.
